# A Review of Different Vaccines and Strategies to Combat COVID-19

**DOI:** 10.3390/vaccines10050737

**Published:** 2022-05-09

**Authors:** Srinivasan Sabitha, Nagarajan Shobana, Pandurangan Prakash, Sathiyamoorthy Padmanaban, Mahendran Sathiyashree, Subramanian Saigeetha, Srikumar Chakravarthi, Saji Uthaman, In-Kyu Park, Antony V. Samrot

**Affiliations:** 1School of Bio and Chemical Engineering, Sathyabama Institute of Science and Technology, Sholinganallur, Rajiv Gandhi Salai, Chennai 600119, India; sabithasrinivasan12@gmail.com (S.S.); shobanan1993@gmail.com (N.S.); kpprakashmtech@gmail.com (P.P.); sathiyasree02@gmail.com (M.S.); 2Department of Biomedical Sciences, Chonnam National University Medical School, Gwangju 58128, Korea; ayanaravinth96@gmail.com; 3Biomedical Science Graduate Program (BMSGP), Chonnam National University, Gwangju 58128, Korea; 4Department of Biotechnology, School of Biosciences and Technology, Vellore Institute of Technology, Vellore 632014, India; rajivarsha2000@gmail.com; 5School of Bioscience, Faculty of Medicine, Bioscience and Nursing, MAHSA University, Jalan SP2, Bandar Saujana Putra, Jenjarom 42610, Malaysia; srikumar@mahsa.edu.my; 6Department of Chemical and Biological Engineering, Iowa State University, Ames, IA 50011, USA; 7Centre for Materials Engineering and Regenerative Medicine, Bharath Institute of Higher Education and Research, Selaiyur 600073, India

**Keywords:** SARS-CoV-2 infection, polymeric nanovaccine, humoral immunity, cell-mediated immunity

## Abstract

In December 2019, an unknown viral infection emerged and quickly spread worldwide, resulting in a global pandemic. This novel virus caused severe pneumonia and acute respiratory distress syndrome caused by Severe Acute Respiratory Syndrome Coronavirus 2 (SARS-CoV-2). It has caused 6.25 millions of deaths worldwide and remains a major concern for health, society, and the economy. As vaccination is one of the most efficient ways to combat this pandemic, different vaccines were developed in a short period. This review article discusses how coronavirus affected the top nations of the world and the vaccines being used for the prevention. Amongst the vaccines, some vaccines have already been approved, and some have been involved in clinical studies. The article also provides insight into different COVID-19 vaccine platforms, their preparation, working, efficacy, and side effects.

## 1. Introduction

According to the World Health Organization (WHO), unknown pneumonia cases were discovered in the central Chinese city of Wuhan in December 2019, and at the end of December 2019, the causative agent was found to be coronavirus [1]. It quickly spread worldwide, resulting in a global pandemic [2]. The virus was given the name Severe Acute Respiratory Syndrome Coronavirus 2 (SARS-CoV-2) by the International Committee on Virus Taxonomy [3], and the WHO named it Coronavirus Disease 2019 (COVID-19) [4]. The virus belongs to the family *Coronaviridae* and has a single-stranded positive-sense RNA genome between 26 and 32 kb in length and comprises 6 to 11 open reading frames [5]. It has four genera, namely α, β, γ, and δ coronaviruses [6]. Spike protein (S), envelope (E), membrane (M), and nucleocapsid (N) are the four structural proteins encoded by coronavirus [7]. The primary target of this virus is lung epithelial cells, and it enters the host cell via angiotensin-converting enzyme 2 (ACE2) receptors [8,9]. Kaur and Gupta [10] explained the structure and genomic organization of coronavirus. SARS-CoV-2 primarily spreads via respiratory droplets or close contact with an infected person. It can also be transmitted through fomites, such as floors, computers, plastics, steel, etc., previously touched by an infected person [11]. The average incubation period of the virus is about five days, although it can range from 1 to 14 days, and within 12.5 days of contact, 95% of patients might experience mild to severe symptoms [12]. The common symptoms of COVID-19 disease are fatigue, fever, cough, myalgia, headache, diarrhea, loss of taste or smell, shortness of breath, and a sore throat. Pneumonia can worsen over time, causing significant respiratory distress in the patient [13]. RT-PCR is used to detect SARS-CoV-2 RNA and quantify the viral load (cycle threshold) [14] by examining the viral RNA levels in pharyngeal and anal swabs [15]. Individuals can prevent SARS-CoV-2 infection by self-isolation, social distancing, personal cleanliness, wearing masks, maintaining well-ventilated rooms [16] and decontaminating floors using disinfectants like sodium hypochlorite [17]. A schematic overview of coronavirus, its prevention, transmission, signs and symptoms, and lifecycle is shown in Figure 1. Vaccination is an effective strategy to help an individual’s immunization against SARS -CoV 2, a virus previously unknown by the human immune system. However, vaccines for the current outbreak must be produced rapidly and be scalable. Vaccines are expected to evoke robust immune responses and longer immune memory responses [2]. The preparation, working, efficacy, and side effects of different COVID-19 vaccines are discussed in this review.

## 2. Background (Origin and Spread of COVID-19)

The outbreak began on 12 December 2019, in Wuhan City in Hubei Province in China, when a cluster of patients reported to a local hospital with fever, cough, dyspnea, and atypical pneumonia. The wholesale Huanan seafood market in Wuhan city in China, where live animals were traded, was home to many of the earliest incidents reported [3,19]. When samples from patients admitted in the intensive care unit (ICU) of Wuhan Jinyintan Hospital in the Dongxihu District of Wuhan were submitted to the Wuhan Institute of Virology, and the samples were tested positive for CoV using Pan-CoV PCR primers [20]. By sequencing, the agent was found to be β-coronavirus and showed similarities to MERS-CoV and SARS-CoV. The transmission of SARS and MERS mostly occurs within small groups, whereas SARS-CoV-2 spreads to the overall community [21]. In the first week of January, China’s government announced that the atypical pneumonia was caused by SARS-CoV2, not the SARS or MERS coronaviruses [22]. It quickly spread across the world, causing the WHO to proclaim it a worldwide pandemic on 11 March 2020 [23]. The R_0_ value (reproductive number) for SARS-CoV-2, a measure of an infectious agent’s contagiousness, was discovered to be 2–3, which later reduced due to the implementation of lockdowns [10,24,25]. COVID-19 puts patients with underlying morbidities, such as chronic lung disease, high blood pressure, severe heart disease, diabetes, and obesity, at a higher risk of mortality [26]. Millions of people have died from the infection, and it has had a major impact on daily lives and economies.

## 3. COVID-19 Cases across the World

COVID-19 has impacted almost all countries. Figure 2 depicts a graphical representation of the COVID-19 infections, deaths, and recovered cases in 10 different countries (as of 28 February 2022). The United States (US) ranked first in terms of infection and fatality cases among countries such as India, Brazil, Russia, France, the United Kingdom, China, Malaysia, Italy, and Germany, with 80,560,293 and 972,930 infections and mortalities, respectively, followed by India with 42,916,117 and 513,756 infections and mortalities, respectively (Source: COVID Live update). The second wave of coronavirus wreaked havoc in the US, the United Kingdom, and India, with many nations implementing strict lockdown and quarantine policies to minimize and control the spread of COVID-19, where businesses, educational institutions, and non-governmental organizations were also closed [27,28]. These lockdowns had to be maintained during the pandemic [29]. Traditional medicines have been used to treat and prevent COVID-19 in China, South Korea, and India [30]. The infection rate in the second wave was found to be 3.78 times higher than in the first wave [31]. A global effort was exerted to combat the pandemic, with many organizations raising donations and humanitarian relief funds. Diagnostic kits, drugs, and vaccines have all been developed and approved. Apart from affecting people’s physical health, the pandemic also significantly impacted their mental health, education, economic status, and lifestyle [32,33].

## 4. Vaccine Strategies

While the pandemic has resulted in high morbidity and mortality, there was emergence of new variants (Delta, Omicron), which is also be needed to be controlled. Vaccination is the most important way to protect people worldwide from COVID-19 as SARS-CoV-2 is highly infectious and affects people globally [34]. The term “vaccine” refers to a biological product that triggers an immune response and protects against illness when a pathogen enters the body, and vaccination refers to the procedure of administering the vaccine [35,36]. Vaccination’s primary goal is to prevent disease and reduce death or disability, but it does not always protect against infection. Vaccine development is a complex process that takes an average of 15 years. The development of vaccines against COVID-19 has been substantially aided by progress in genomic sequencing and technology [23]. Because of the flexibility and speed of the various vaccine technologies, vaccines can be developed quickly [37]. According to Bartsch et al. [38], a vaccine must have at least 70% efficacy to avoid an epidemic and 80% efficacy to largely eradicate an epidemic. In addition to efficacy, the delivery method, acceptance by the community, its ability to reduce infection, duration, and safety must be considered [39]. Spike protein (S1 subunit) is targeted by the SARS vaccine candidates to stimulate neutralizing antibodies and protective immunity [40]. The different types of COVID-19 vaccines include inactivated viruses, mRNA-based vaccines, live-attenuated vaccines, DNA vaccines, viral-vector-based vaccines, and vaccine against protein subunits, inactivated virus, mRNA-based vaccine, live-attenuated vaccine, DNA vaccines, viral-vector-based vaccine, and protein subunits [41]. Every vaccine platform has its own set of characteristics that differentiates it from others and could have important implications for the vaccine’s efficacy, induced protection period, and safety [42]. According to the WHO COVID-19 vaccine tracker, there are now 146 vaccines in clinical development and 195 in pre-clinical development (as of February 2022). According to the WHO Coronavirus (COVID-19) Dashboard, a total of 10.7 billion vaccine doses have been administered across the globe (as of 28 February 2022). Knowing the type and length of the immune response following COVID-19 immunization could help researchers determine the immunological mechanisms involved in disease protection [43]. If any side effects occur following immunization, all individuals should seek medical assistance immediately.

### 4.1. mRNA-Based Vaccines

mRNA-based vaccines provide a rapid immune response through the quick translation of antigen in the target cell but are extremely susceptible to degradation because the single-stranded mRNA structure can be quickly degraded by RNases [44]. By utilizing nanoparticle-based complexing agents, such as lipids and polymers, the ability of mRNA vaccines to deliver mRNA to the cytoplasm for translation can be improved [45]. mRNA vaccines could benefit vaccine development since they can mimic natural infections and stimulate the immune system to inhibit their spread [46]. Some of the mRNA-based vaccines for treating COVID-19 are the Comirnaty^®^ (BNT162b2) and Moderna vaccines (mRNA-1273) [47].

#### 4.1.1. Comirnaty^®^ Vaccine-Pfizer-BioNTech

Pfizer Inc., (New York, NY, USA) and BioNTech, (Mainz, Germany) developed the Comirnaty^®^ vaccine, which became the first vaccine approved and authorized for use by the US Food and Drug Administration (FDA). It is an mRNA-based vaccine that is encapsulated in lipid nanoparticles, which encodes for the full-length spike of the SARS-CoV-2 virus and is modified with two proline mutations (P2S) that are locked into the prefusion conformation so they can evoke antibody responses that neutralize the virus [48,49]. The FDA granted the first emergency use authorization (EUA) of this vaccine for COVID-19 prevention in December 2020 and approval on 23 August 2021, and it is now marketed as Comirnaty [50]. It offers 95% protection against COVID-19 [51]. The vaccine vials must be stored at −70 °C and have a shelf-life of up to 6 months [52]. Two doses (0.3 mL each) are administered intramuscularly, 3 weeks apart in the upper arm after mixing with saline [53]. Heat, pain, redness, swelling, fatigue, fever, vomiting, headache, myalgia, and diarrhea were the most common side effects after administration, and an anaphylaxis reaction has also been reported [54,55]. For people aged over 18 years, a booster dose is given at least 6 months after the second vaccination. The Comirnaty^®^ vaccine is also extremely effective at preventing the spread of the B.1.351 variant [56].

#### 4.1.2. mRNA-1273 Vaccine-Moderna

Moderna developed the mRNA-1273 vaccine, which is a lipid nanoparticle-encapsulated mRNA vaccine (Figure 3). It encodes for the coronavirus spike protein, which is prefusion-stabilized. The mRNA-1273 vaccine received EUA from the FDA, making it the second FDA-approved vaccine against COVID-19 infection for individuals 18 years of age and older [57]. Corbett et al. [58] presented a detailed method of preparation, expression, and analysis of the mRNA vaccine. It is a sterile injection with a 0.5 mg/mL dosage, and saline is used as the diluent [59]. The mRNA-1273 vaccine has a 94.1% efficacy with two doses of 0.5 mL each. It is given intramuscularly 28 days apart to individuals 18 years of age and older and stored at −20 °C [60]. Fatigue, vomiting, fever, myalgia, arthralgia, joint pain, swollen lymph nodes, and headaches are some of its side effects [61]. The FDA revised the Moderna COVID-19 Vaccine EUA on 20 October 2021 to allow a single booster dose to be given at least 6 months following the primary doses. This vaccine is 81.6% and 95.7% effective against B.1.1.7 and B.1.351 variants, respectively. It is manufactured by ModernaTX, Inc. Cambridge, Massachusetts and marketed as the Moderna COVID-19 Vaccine [62].

### 4.2. Inactivated Virus Vaccines

Growing a virus in culture and inactivating it with a chemical that allows for the stable expression of native antigenic epitopes can be used to make an inactivated virus vaccine [63]. Dead viruses are used as the immunogen in inactivated vaccine to stimulate immune responses. Designing inactivated viruses is probably safer, but it usually only provides short-term immunity and necessitates frequent boosters [64]. The vaccines developed by Sinovac, Sinopharm, and Bharat Biotech are discussed below.

### 4.2.1. CoronaVac Vaccine-Sinovac Biotech

CoronaVac is an inactivated virus vaccine developed by Sinovac Biotech, (Beijing, China) and is marketed as CoronaVac or PiCoVacc. It was developed by cultivating SARS-CoV 2 into African green monkey kidney cells (Vero cells) and then inactivating them with β-propiolactone, which was then mixed with aluminum hydroxide adjuvant. It does not need to be frozen, and it can be stably stored at 2–8 °C for up to 3 years. It can be mixed and matched with other vaccines, such as COVID-19 mRNA vaccines (Pfizer or Moderna) or COVID-19 vector vaccines (COVISHIELD or Janssen) [65,66]. It was granted marketing authorization on 6 February 2021, by the China National Medical Products Administration (NMPA). It has shown 83.5% efficacy after 14 days or more from the second dose [67]. It involves two 0.5 mL administrations in the deltoid muscle at 2–4 week intervals between the first and second dose. The side effects include headaches, fatigue, muscle pain, and vomiting.

### 4.2.2. BBIBP-CorV Vaccine-Sinopharm

The BBIBP-CorV vaccine is an inactivated virus vaccine developed by Sinopharm in collaboration with the Beijing Institute of Biological Products and the Chinese Center for Disease Control and Prevention. It was developed by culturing the virus with Vero cells and further inactivation using β-propiolactone. Aluminum hydroxide was used as the adjuvant, as in the inactivated polio vaccine. Each prefilled syringe contains a 0.5 mL dose in sterile phosphate-buffered saline [68,69]. The China National Medical Products Administration authorized its use on 31 December 2020. Wang et al. [70] proved this vaccine to be highly protective against COVID-19 with good genetic stability. It is recommended for people aged 18 and older and given in two doses in the deltoid muscle 3 to 4 weeks apart [71]. The overall efficacy was proven to be 79%, and the vaccine can be stored at 2 to 8 °C [69]. Pain at the injection site, fatigue, soreness, lethargy, and headaches were the most prevalent reported side effects [72].

### 4.2.3. Covaxin (BBV152) Vaccine-Bharat Biotech

Bharat Biotech developed Covaxin, India’s indigenous COVID-19 vaccine, in partnership with the Indian Council of Medical Research-National Institute of Virology (ICMR-NIV). On 3 January 2021, the Drugs Controller General of India (DCGI) approved the use of Covaxin to prevent COVID-19 [73]. Covaxin was constructed by inactivating whole-virion with β-propiolactone [2]. The vaccine is only available to people over 18 years of age and must be administered in two doses of 0.5 mL each, with a 28 day interval between the first and second doses. It does not require sub-zero storage, does not require reconstitution, and is available in multi-dose vials. The drug is administered intramuscularly and is stable at temperatures ranging from 2 to 8 °C. It has also been shown to effective against the Delta (B.1.617.2) and Beta (B.1.351) variants. It has an efficacy of 77.8%, with side effects such as injection site pain, itching, headache, fever, body aches, nausea, and vomiting [74].

### 4.3. Viral Vector-Based Vaccines

A viral vector-based vaccine involves cloning the gene that encodes for the pathogenic antigen into viral vectors that are either non-replicating or replicating [75]. A vaccine derived from viral vectors can be manufactured quickly without an adjuvant [40]. Covidshield, Sputnik V, and Ad26.COV2.S have been made for preventing COVID-19 infections by utilizing viral-based technology.

#### 4.3.1. CoviShield (ChAdOx1 nCoV) Vaccine-AstraZeneca

Covishield (ChAdOx1 nCoV-19) was developed by Oxford University in the United Kingdom and the British-Swedish company AstraZeneca [76]. It is commercialized under the names Vaxzevria and Covishield. It is a chimpanzee adenovirus-vector vaccine that encodes for a full-length S-protein of SARS-CoV2 and is produced in genetically modified human embryonic kidney (HEK) 293 cells [77,78]. The Serum Institute of India at Pune is one of its manufacturing sites. It is administered intramuscularly in the upper arm for people above 18 years of age in two shots of 0.5 mL each, 3 months apart, and has shown 90% efficacy. Fatigue, headache, fever, vomiting, chills, diarrhea, and myalgia were the common side effects after vaccination, and anaphylaxis has also been reported [79]. Schultz [80] reported that the patients developed severe thrombosis and thrombocytopenia after receiving the first dose. The five healthcare workers aged 32 to 54 years revealed venous thrombosis and thrombocytopenia 7 to 10 days after the initial dose. This shows that the thrombocytopenia was caused by a rare vaccine-related type of spontaneous heparin-induced thrombocytopenia. It is 81% effective against the Alpha variant (lineage B.1.1.7) and 61% effective against the Delta variant (lineage B.1.617.2).

#### 4.3.2. Sputnik V (Gam-COVID-Vac) Vaccine-Gamaleya

The Russian Defense Ministry and the Gamaleya Research Institute of Epidemiology and Microbiology developed the Sputnik V-Gam-COVID-Vac vaccine, which is generated from rAd type 26 (rAd26) and rAd type 5 (rAd5) and contains the entire S-protein of SARS-CoV2 [81,82]. On 11 August 2020, the Russian Federation’s Ministry of Health approved this, the world’s first registered combination vector vaccination, for administration [83]. It is marketed under the trade name Sputnik V. The name of the vaccine was chosen to honor the launch of the world’s first artificial Earth satellite in 1957. It is administered in two 0.5-mL doses into the deltoid muscle, 21 days apart, and stored at −18.5 °C (liquid form) and 2–8 °C (dry form) [84]. The effectiveness of the vaccine for symptomatic COVID-19 is 91.6%, preventing the disease completely. Some of the side effects are body soreness, joint pain, chills, injection site pain, redness, fever, and headache [85].

#### 4.3.3. JNJ-78436735 (Ad26.COV2.S) Vaccine-Janssen

Janssen Vaccines in Leiden, Netherlands, collaborated with Johnson & Johnson to create the Ad26.COV2.S vaccine. It consists of a recombinant, non-replicating adenovirus serotype 26 vector vaccine and SARS-stabilized CoV-2 S-protein [86]. It is marketed under the name Janssen COVID-19. Citrate, sodium chloride, hydroxypropyl-cyclodextrin, polysorbate, and ethanol at 6.2 pH are utilized to prepare the formulation buffer, and 0.9% sodium chloride is employed as the placebo [87]. It was authorized by the US FDA on 27 February 2021 after it was shown to be safe and immunogenic in humans and nonhuman primates [88]. For those aged 18 years and up, a single dose of 0.5 mL is delivered intramuscularly, and the vaccine does not need to be frozen. The WHO recommends a second dose for immunocompromised people aged 18 years and older, 1 to 3 months after the first dose. Efficacy was found to be 66.9%, with some side effects, including fatigue, headache, myalgia, fever, chills, nausea, redness, and diarrhea [89,90].

### 4.4. Protein Subunit

A vaccine based on protein subunits consists of antigenic pathogen fragments and is generally made using recombinant protein or synthetic peptides. They are safe, effective, target-specific, neutralize the antigen, and increase immunogenicity [91,92]. However, they require optimized adjuvants for stronger and long-lasting immune responses. Subunit vaccines can be designed to concentrate the immune response on neutralizing epitopes, thereby preventing the development of non-neutralizing antibodies that could promote disease development [93]. NVX-CoV2373 is one such protein subunit-based vaccine for treating COVID-19.

#### NVX-CoV2373-Novavax

The NVX-CoV2373 vaccine, manufactured by Novavax and the Coalition for Epidemic Preparedness Innovations (CEPI), uses genetically engineered long S-protein instead of wild-type S-protein and contains Novavax’s proprietary Matrix M adjuvant, which is a combination of lipid molecules and saponins for stabilizing the structure and enhancing immunity [94,95]. It is sold under the brand names Nuvaxovid and Covovax [96]. Takeda Pharmaceutical Company, Japan, and the Serum Institute of India are its manufacturing sites. This vaccine is eligible for administration to people older than 18 years and is delivered in two doses (0.5 mL each) at a time interval of 21 days. The efficacy was shown to be about 89.7 % against COVID-19 disease, and it had 96.4% efficacy against non- B.1.1.7 variants. The vaccine can be stored at 2–8 °C. Injection-site tenderness, fatigue, chills, fever, redness, headache, and muscle pain are some of the common side effects of this vaccine [97,98].

### 4.5. DNA-Based Vaccines

The gene coding for immunogenic antigens is delivered to host cells by plasmid DNA in a DNA vaccine [99]. It is possible to quickly make DNA vaccines using synthetic methods instead of cultures or fermentation [100]. In contrast to other vaccine technologies, there are no concerns about Th2-skewed immunity with DNA vaccines [101]. Because DNA is stable, it can elicit humoral and cellular immune responses and has higher productivity, and stability and longer shelf-life, and storage [102]. Since balanced T-cell response is required for an effective immune response, immune response skewing has been associated with the emergence of pathologic responses, such as that seen in Respiratory syncytial virus (RSV) infection, which has enhanced Th2 and Th17 responses with attenuated the Th1 responses [103]. Th2-skewed immune response is especially prominent in the lungs, resulting in severe bronchiolitis and asthma development, as well as respiratory dysfunction [104]. Th2-skewed immunity increases the risk of reinfection and induces the thickening of the epithelium [105]. These vaccines are generally safe, but their immunogenicity tends to be lower. As a result, they often require sequence optimization, the use of adjuvants, or the use of combined immunization techniques [106]. The major problem with DNA-based vaccines is that foreign genetic information can be transfected by integrating into the host’s chromosomes and can cause mutations [45]. Zyvox-D Vaccine is one of the DNA-based vaccines for treating COVID-19 disease.

#### ZyCov-D Vaccine-Cadila Healthcare

The Ahmedabad-based pharmaceutical company Cadila Healthcare developed Zyvox-D Vaccine in India in association with the Biotechnology Industry Research Assistance Council and is the world’s first DNA plasmid-based vaccine. It is made up of the DNA plasmid vector pVAX1 that expresses S-protein SARS-CoV2 and IgE signaling peptide [107]. It is India’s first needle-free COVID-19 vaccination and is licensed for emergency usage. It is administered with a disposable needle-free injector, which delivers a narrow stream of the fluid intradermally. It can be stored at 25 °C for at least 3 months or at −2 to 8 °C for transportation. The ZyCoV-D vaccine is recommended for young people in the 12–18-year age group. It is given in three doses of 0.5 mL each at 28-day intervals and has an efficacy rate of 66.6%. Body aches, nausea, redness, low-grade fever, and vomiting are the common side effects [108]. Table 1 shows a comparison between different COVID-19 vaccines in terms of preparation, efficacy, side effects, and dosage and Figure 3 shows the immune response expected to be created in host system [109,110,111]. 

## 5. Delta and Omicron Variants and the Current Scenario

The Delta variant of coronavirus, namely the B.1.617.2 lineage, was first reported in India in December 2020 [112] and was classified as a variant of concern (VOC) by the WHO due to its rapid spread [113]. The severity of the infection was the worst, and the vaccine’s effectiveness was reduced [114]. The doubling time of this Delta variant is 3.5 to 16 days [115]. More than 15 mutations have been found in this lineage, and six main mutations related to amino acid changes, including D614G, L452R, T478K, P681R, D950N, and T19R del157/158, were found [116]. It is the most highly transmissible variant with a transmission rate more than 60% higher than that of the Alpha variant (B.1.1.7), and it spread to at least 62 countries [117,118]. Transmissibility was increased by genomic alterations, such as mutations in the N-terminal antigenic supersite, the polybasic furin cleavage site, and on the receptor-binding domain (RBD) of the S-gene, which led to the increased binding affinity of the virus to the host through the ACE2 receptor and enhanced viral entry [119,120]. This variant was responsible for the worst second wave of COVID-19 in India, beginning in April 2021. It was first reported in the US in March 2021. It was first recognized in Guangzhou, China Guangdong in May 2021 [121,122]. The symptoms of this variant are different from the other variants and are more severe than those of the previously reported variants. They include high-grade fever, shortness of breath, headache, fatigue, runny nose, sore throat, and loss of taste or smell [123]. Bernal et al. [124] reported vaccine effectiveness of 88% using BNT162b2 (Pfizer–BioNTech) and 67 % with ChAdOx1 AZD1222 (Covishield) vaccine against the Delta variant. Ad26.COV2.S (Johnson & Johnson vaccine) was effective against the Delta variant with neutralizing activity. The BBV152 (Covaxin) vaccine was also reported to be 65.2% effective against the Delta variant with about a three-fold drop in antibody titer [74,125]. Fully vaccinated individuals are believed to have a 50 to 60% reduced risk of infection with the Delta variant [126].

Recently, a novel variety termed Omicron (B.1.1.529) was discovered with a higher transmission rate than any other strain discovered before considering the number of mutations. It has a 70-fold faster spread rate than the Delta variant, and the WHO believes this variant could be more damaging than the previous variants due to its mutable nature [127]. SARS-CoV-2 Omicron was believed to cause the fourth wave of the COVID-19 pandemic, following D614G, Beta, Gamma, and Delta VOCs [128]. The Omicron strain was initially discovered in South Africa November 2021, and it has since surpassed the Delta variety as the most common strain in most countries [129]. On 26 November 2021, the WHO classified it as a VOC. More than 60 substitutions, insertions, and deletions have been identified in the structure of the Omicron variant [130]. About 30 amino acid alterations, one small insertion, and three short deletions were found in the spike glycoprotein, with 15 mutations occurring in the receptor-binding domain (RBD) [131]. People in a total of 170 countries have been infected, including the UK, USA, Denmark, Germany, Canada, India, France, Switzerland, Japan, and Australia. With almost 517,000 cases as of February 2022, the United Kingdom has the highest SARS-CoV-2 Omicron variant infection rate [132,133]. This variant can be detected via RT-PCR, and its doubling time is 1.2 days [134]. In contrast to other COVID-19 infections, patients with this variant reported only mild to moderate symptoms, such as a cold, fever, night sweats, cough, tiredness, headache, fatigue, sore throat, runny nose, and loss of taste and smell [135]. It is believed to be 91% less fatal than the Delta variant. Despite the high number of infections, hospitalizations and mortality are declining [136]. BA.2 is a subvariant of the Omicron strain that is 1.5 times more transmissible than the original strain. Studies overwhelmingly agree that COVID-19 severity can be decreased due to vaccination, with growing concerns over the spread of the Omicron variant. Whatever the variant may be, it is advisable to take precautions and follow preventive measures given by the government. A precautionary shot/booster dose has been introduced in India amid a surge in infections. Twenty-five weeks after the second dose, the third or booster dose has resulted in a 57% reduction in the likelihood of symptomatic S-gene-negative infections [137].

## 6. Role of Antioxidants in COVID-19 Prevention

According to recent studies, SARS-CoV-2 infection causes both direct and indirect oxidative stress by increasing reactive oxygen species (ROS) and decreasing host antioxidant defense mediated by nuclear factor (erythroid-derived 2)-like 2 (nuclear factor erythroid 2-related factor 2) [138]. The infection results in reduced glutathione (GSH), which protects against oxidant damage. When the body is subjected to high levels of ROS, such as superoxide (O^2−^), hydroxyl (OH^−^), and hydrogen peroxide (H_2_O_2_), it experiences oxidative stress, consequently hampering the antioxidant system, consisting of enzymes such as catalase, glutathione peroxidase, albumin superoxide dismutase, and bilirubin [139]. The term “antioxidant” refers to a molecule that prevents oxygen consumption. It has radical scavenging properties, which donate an electron to the free radical and neutralize it, thus limiting the harmful effects and preventing cell damage [140,141]. Natural and synthetic antioxidants can neutralize the effects of oxidants in the following ways: (i) direct neutralization of ROS; (ii) inhibition of oxidant-generating enzymes, such as xanthine oxidase; (iii) activation of ROS-metabolizing enzymes, such as glutathione peroxidase catalase, superoxide dismutase, and (iv) regulation of redox-sensitive transcription factors. As a result, antioxidant agents can inhibit the development of ROS and attenuate their negative effects [142]. Additionally, to compensate for an antioxidant shortage, the body needs exogenous antioxidants obtained from food and nutritional supplements. The use of antioxidant-rich agents and food supplements, such as flavonoids (quercetin), melatonin, sodium selenite, N-acetylcysteine, vitamin A, vitamin C, Nrf2 activators (e.g., curcumin), hydroxychloroquine, vitamin D, cinnamaldehyde (cinnamon), allicin (garlic), piperine (black pepper), selenium (cabbage, corn, onion, garlic, and broccoli), lactoferrin, carotenoids, and polyphenols, has been tested or proposed to reduce the risk of infection with SARS-CoV-2 [138,139,142,143,144,145]. Ebselen (2-phenyl-1,2-benzoisoselenazol-3(2H)-one), an organoselenium compound with hydroperoxide- and peroxynitrite-reducing activity, acts by inhibiting the main viral protease, Mpro. It has proven potential beneficial effects against COVID-19 [146,147,148,149,150].

These antioxidants help in the prevention of diseases in the following ways: (i) by acting as a contributor to cytokine downregulation and lowering reactive oxidative stress [151]; (ii) by regulating the elements responsible for an antioxidant response like ARE-dependent genes [152]; (iii) by quenching ROS, such as singlet oxygen, and lipid peroxides present in the bilipid layer of the cell membrane [143,153]; (iv) by reducing the likelihood of cytokine storm [154], and (v) by restoring the antioxidant capacity of the host, thus reducing cell damage and the aggregation of platelets (Figure 4) [155,156]. Traditional Siddha formulations may also contribute in managing respiratory symptoms in patients with COVID-19 [157].

## 7. Conclusions

COVID-19 has caused chaos worldwide. The development of vaccinations has given people optimism that COVID-19 can be controlled. Vaccination protects not just the person who receives it, but also protects the society indirectly by reducing hospitalizations and admissions in intensive care units. Vaccinated people are less likely to have long-term COVID-19 symptoms. Vaccinated people are less likely to acquire acute symptoms and are more likely to develop milder, less frequent symptoms than the unvaccinated people. Unvaccinated people encounter a variety of symptoms, which can include fever, tiredness, headache, cough, shortness of breath, and, in some cases, low blood oxygen levels. These symptoms may necessitate hospitalization, intubation, and, in the most severe cases, death. Prevention is better than a cure, so get vaccinated and always adhere to COVID-19 guidelines. Wear a mask, wash your hands frequently, keep a safe distance from others, and consume nutritious meals. Stay safe, stay healthy.

## Figures and Tables

**Figure 1 vaccines-10-00737-f001:**
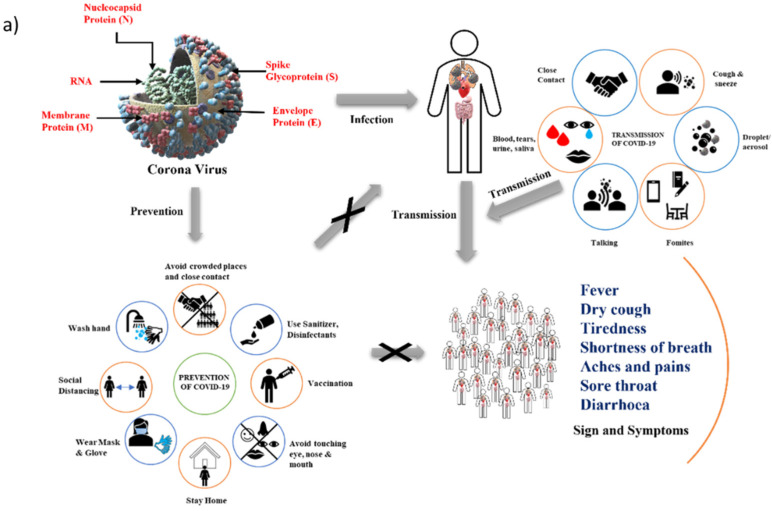
Schematic overview of COVID-19 transmission and lifecycle: (**a**) Prevention, transmission, signs and symptoms, and (**b**) lifecycle of coronavirus (Inspired from Liu et al. [18]).

**Figure 2 vaccines-10-00737-f002:**
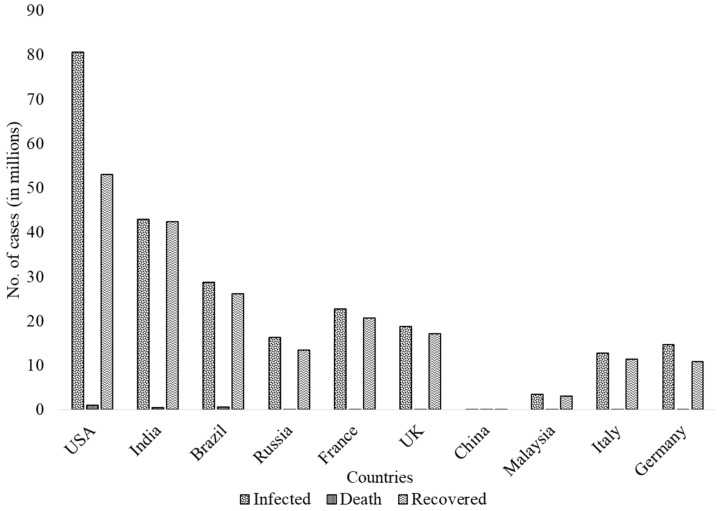
Graphical representation of COVID-19 infections, deaths, and recovered cases in 10 different countries (as of 28 February 2022).

**Figure 3 vaccines-10-00737-f003:**
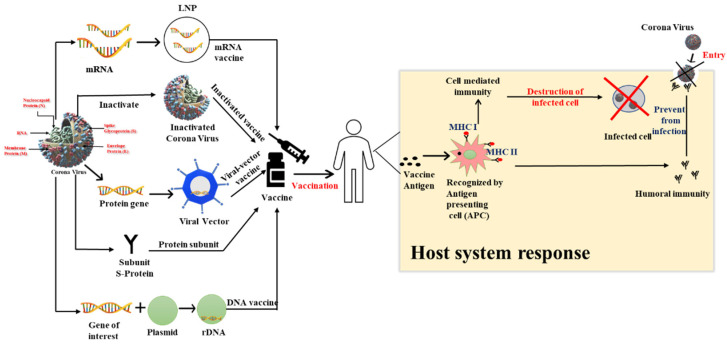
Vaccine strategies and immune response.

**Figure 4 vaccines-10-00737-f004:**
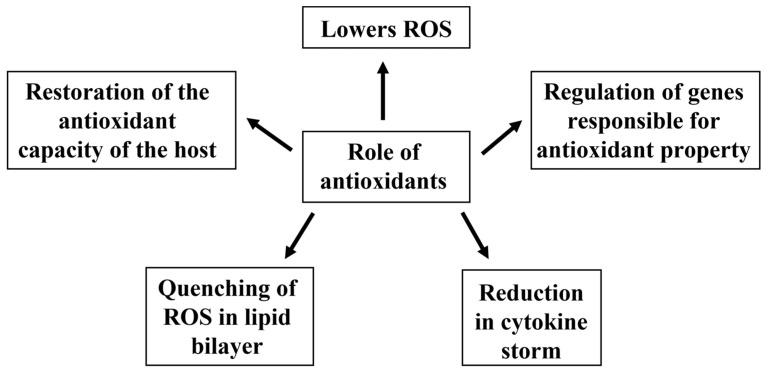
Prevention of diseases using antioxidants.

**Table 1 vaccines-10-00737-t001:** Comparison of different COVID-19 vaccines.

TYPE	mRNA Vaccine		Inactivated Virus Vaccine			Viral Vector Vaccine			Protein Subunit	DNA Based Vaccine
VACCINE NAME	Comirmaty^®^	mRNA-1273	CoronaVac	BBIBP-CorV	Covaxin	Covishield	Sputnik V	Janssen Vaccines	Nuvaxovid	ZyCoV-D
**MANUFACTURER**	Pfizer Inc & BioNTech	Moderna	Sinovac	Sinopharm	Bharat Biotech	AstraZeneca/ University of Oxford	Gamaleya	Janssen	Novavax	Cadila Healthcare
**PREPARATION**	S-Protein (SARS-CoV2) + P2 S → LNP	S-Protein → LNP	Inactivated virus created from African green monkey kidney cells (Vero cells)	β-propiolactone (inactivate COV) + adjuvant (Al(OH)_3_) + Vero cell	Whole-virion inactivated SARSCoV-2 antigen	S-protein (SARS-CoV-2) + chimpanzee adenovirus vector	rAd26 & rAd5-both carry the gene for S-protein (SARS-CoV-2)	Non-replicating adenovirus serotype 26 +S-protein (SARS-CoV-2)	Long S-Protein (SARS-CoV-2) + M adjuvant	S-protein (SARS-CoV-2) + pVAX1
**DOSAGE**	2 doses, 0.3 mL each 21 days apart	2 doses, 0.5 mL each 28 days apart	2 doses, 0.5 mL each 28 days apart	2 doses, 0.5 mL each 21 days apart	2 doses, 0.5 mL each 28 days apart	2 doses, 0.5 mL each 3 months apart	2 doses, 0.5 mL each 21 days apart	1 dose 0.5 mL	2 doses, 0.5 mL each 21 days apart	3 doses, 0.5 mL each 28 days apart
**AGE GROUP**	12 and older	18 and older	18 and above	18–59	18 and above	18 and older	18 and above	18 and older	18 and above	12 and above
**% EFFICACY**	95%	94.1%	83.5%	79%	77.8%	90%	91.6%	66.9%	89.7%	66.6%
**SIDE EFFECTS**	Headache, Fatigue, chills, Muscle and joint pain	Fatigue, Myalgia, Arthralgia, Headache	Headache, Fatigue, Muscle pain, Vomiting	Pain at the injection site, Fatigue, Lethargy, Headache, Tenderness	Injection site pain, Itching Headache, Fever, body ache, Nausea, Vomiting	Headache, Fever, Dizziness Nausea Vomiting, Myalgia.	Body pain, Injection site pain, Headache and Fatigue	Fatigue, Headache, Myalgia, Fever, Chills, Nausea, Diarrhoea	Injection-site tenderness, Fatigue, Headache, Muscle pain	Body ache Nausea Vomiting, Fever, Chills
**EFFICACY AGAINST DELTA VARIANT**	88%	86.7%	59%	Not much information available	65.2%	65.2%	83%	71%	Not much information available	66.6%
**REFERENCES**	[50,51,53]	[61,62]	[66,67]	[69,71]	[74]	[79]	[82,84,85]	[89]	[97,98]	[108]

## Data Availability

Not applicable.

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
