# Peer review of "A Review of Different Vaccines and Strategies to Combat COVID-19"

_vaccines, 2022, doi:10.3390/vaccines10050737_

Round 1

Reviewer 1 Report

The manuscript exposes a review of prevention in COVID-19 pandemics, with a main important part of vaccination implementation. The article is a review with a important bibliography, which could allow a good overview of the prevention intervention in public health and its impact on the improvement of the epidemics.

The title presents the "review  of different vaccines and strategies to combat COVID_19";  However the focus is mainly done on vaccines, which is of particular interest and well proposed.

Hence the subject and the body text are of interest for the readers of Vaccine, but the messages are a little to diluted by the introduction concerning the prevention of SARS-COV2 and the end of the review with the point 6 anti-oxydants.

Indeed, if the work could be suitable for publication, it must be reviewed with some editing.

On my opinion, the introduction could be reduce a lot by referring to the figure 1 and synthetic the second part of the introduction by a half. On the same way, the second part concerning the background and the third one concerning the spread across the world could be gather together in one large chapter of the evolution of the pandemics and be shorten too.

In the background, the Estimation of the R0 could be enhance with other publication like 

Salje, H. et al. Estimating the burden of SARS-CoV-2 in France. Science 369, 208–211 (2020).

Li, R. et al. Substantial undocumented infection facilitates the rapid dissemination of novel coronavirus (SARS-CoV-2). Science 368, 489–493 (2020).

and other

In fig2, The representation in rates would be more accurate in my opinion

The table 1 is of great interest and could allow shortening the chapter 4

I am not sure that the Fig 3 is necessary; maybe in supplementary file?

The end of the paper must develop more the impact of omicron and the difference in the clinical presentation  between vaccinated people and not vaccinated.

Indeed, omicron is more contagious, But we have first studies showing that it is less severe, and especially when people are fully vaccinated. I think that the review must be the good moment to insist on that with the recent literature!

As previously proposed, I would withdraw the chapter 6and fig4.

The conclusion could be enhance by the chance given by the quick access to vaccination especially in Western countries but even s, and the positive impact of vaccination even with omicron in terms of clinical presentation.. 

The role of vaccine in transmission must be presented with the literature also.

Author Response

We thank the reviewers for their valuable comments and suggestions to improve the manuscript. We have addressed the comments and revised the manuscript according to the comments from the reviewers.

Reviewer 1:

Comments and Suggestions for Authors
The manuscript exposes a review of prevention in COVID-19 pandemics, with a main important part of vaccination implementation. The article is a review with a important bibliography, which could allow a good overview of the prevention intervention in public health and its impact on the improvement of the epidemics.

The title presents the "review  of different vaccines and strategies to combat COVID_19";  However the focus is mainly done on vaccines, which is of particular interest and well proposed.

Hence the subject and the body text are of interest for the readers of Vaccine, but the messages are a little to diluted by the introduction concerning the prevention of SARS-COV2 and the end of the review with the point 6 anti-oxydants.

1. Indeed, if the work could be suitable for publication, it must be reviewed with some editing. On my opinion, the introduction could be reduce a lot by referring to the figure 1 and synthetic the second part of the introduction by a half. On the same way, the second part concerning the background and the third one concerning the spread across the world could be gather together in one large chapter of the evolution of the pandemics and be shorten too.
Response: Thank you for the valuable comment. We have worked on the introduction and made it more interesting to follow through. Regarding chapter 2 and 3, we would like to state the importance of origin and the spread of covid across the globe making it a potential pandemic. This will help the readers to understand the severity of the disease.

  1. In the background, the Estimation of the R0 could be enhance with other publication like
    Salje, H. et al. Estimating the burden of SARS-CoV-2 in France. Science 369, 208–211 (2020).Li, R. et al. Substantial undocumented infection facilitates the rapid dissemination of novel coronavirus (SARS-CoV-2). Science 368, 489–493 (2020) and other In fig2, The representation in rates would be more accurate in my opinion
    Response: Thank you for the comment, here we have added “The R0 value (reproductive number) for SARS-CoV-2, a measure of an infectious agent's contagiousness, was discovered to be 2–3 which later reduced due to the implementation of lockdowns” using Salje H et. al and the other suggested citations. Regarding fig 2, since we have obtained the data regarding the number of cases around the globe, we gave the graphical representation as per the data collected.
  2. The table 1 is of great interest and could allow shortening the chapter 4
    I am not sure that the Fig 3 is necessary; maybe in supplementary file?

Response: Thank you for the valuable comment. Regarding fig 3, we would like to showcase the strategies and immune response hand-in-hand with the text for better understanding and clear picturisation.

4. The end of the paper must develop more the impact of omicron and the difference in the clinical presentation between vaccinated people and not vaccinated. Indeed, omicron is more contagious, But we have first studies showing that it is less severe, and especially when people are fully vaccinated. I think that the review must be the good moment to insist on that with the recent literature!

Response: We thank the reviewer for the comment. We have addressed this comment in the conclusion. We have added “Vaccination protects not just the person who receives it, but also society indirectly by reducing hospitalizations and admissions in intensive care unit Vaccinated people are less likely to have long-term COVID-19 symptoms. Vaccinated people are less likely to acquire acute symptoms and are more likely to develop milder, less frequent symptoms than the unvaccinated people.  Unvaccinated people encounter a variety of symptoms, which can include fever, tiredness, headache, cough, shortness of breath, and, in some cases, low blood oxygen levels. These symptoms may necessitate hospitalization, intubation, and, in the most severe cases, death.” for showcasing the severity of the disease in vaccinated and non-vaccinated individuals.

  1. As previously proposed, I would withdraw the chapter 6 and fig4.

Response: Dear reviewer we would like to showcase the importance of antioxidants in the treatment of COVID and the role it plays in suppressing other diseases too. So allow us to keep the chapter along the image.

  1. The conclusion could be enhance by the chance given by the quick access to vaccination especially in Western countries but even s, and the positive impact of vaccination even with omicron in terms of clinical presentation. The role of vaccine in transmission must be presented with the literature also.

Response: We thank the reviewer for this comment for enhancing our review. We have enhanced it in chapter 5. We are very much thankful for your comments and suggestions for improving our manuscript.

Reviewer 2 Report

General Comments:This is an interesting and well done  concise review about different vaccines and strategies to combat COVID -19 disease . 

Specific Comments: 

1.Pg 2  Line 55 " Real-time reverse transcriptase-polymerase chain reaction 
(RT-PCR) is used to diagnose the severity of COVID-19 "  may be it is better to say :  RT-PCR is used to detect SARS-COV2 RNA and quantify the viral load ( Cycle threshold) .....

2. Pg 2 Line- 60
" vacination is an effective strategy to combat COVID-19 infection"....maybe it is better to say :  vaccination is an effective strategy to help individuals` immunisation against SARS -COV 2,  a virus previously unknown by the human  imune  system.  

Author Response

We thank the reviewers for their valuable comments and suggestions to improve the manuscript. We have addressed the comments and revised the manuscript according to the comments from the reviewers.

Comments and Suggestions for Authors
General Comments:

This is an interesting and well done concise review about different vaccines and strategies to combat COVID -19 disease.

Specific Comments:

1.Pg 2  Line 55 " Real-time reverse transcriptase-polymerase chain reaction
(RT-PCR) is used to diagnose the severity of COVID-19 "  may be it is better to say :  RT-PCR is used to detect SARS-COV2 RNA and quantify the viral load ( Cycle threshold) .....

Response: Thank you for the comment. We have altered the sentence as per the suggestion to “RT-PCR is used to detect SARS-CoV-2 RNA and quantify the viral load (Cycle threshold)”  

  1. Pg 2 Line- 60
    " vaccination is an effective strategy to combat COVID-19 infection"....maybe it is better to say :  vaccination is an effective strategy to help individuals` immunisation against SARS -COV 2,  a virus previously unknown by the human  immune  system.

Response: We thank the reviewer for this comment to improve our manuscript. We have worked on this comment and modified as “Vaccination is an effective strategy to help individual's immunization against SARS -COV 2, a virus previously unknown by the human immune system.”.

We thank the reviewer for the insightful comments to improve the quality of our manuscript

Reviewer 3 Report

Dear Editor

The review by In-Kyu Park and Antony V Samrot deals with strategies to combat COVID19 with total emphasis on vaccination procedures. The review is well written even if sometimes the English can be improved. I highly appreciated the didactic texture and the teaching approach of the manuscript that surely deserves publication in Vaccines after the following points will be addressed:

Major:

  • Authors correctly highlighted the role of antioxidants in the frame of a possible COVID19 treatment without citing some interesting small molecules that are in the pipeline and some others that are in preclinical studies. In the first case Ebselen, the most studied among selenorganic compounds, needs to be cited since it is in phase III of clinical trials (https://clinicaltrials.gov/ct2/show/NCT04484025?term=ebselen&draw=2&rank=10; https://clinicaltrials.gov/ct2/show/NCT04483973?term=ebselen&draw=3&rank=11) this compound needs brief emphasis by citing recent and relevant literature. Here is some suggestions 10.1038/s41467-021-23313-7; 10.1038/s41586-020-2223-y and some recently reported review that authors can select form literature
  • Authors cited quercetin as potential anti-SARSCoV2 agents, without citing related literature, here are some suggestions: 10.1016/j.ijbiomac.2020.07.235; 10.3390/ijms22137048; 10.3390/molecules26196062
  • Table 1. It is really interesting and well-conceived. I would remove the storage since it is not a useful information for readers aside of health authorities who practically will store such vaccines. Rather, I would add information regarding the efficacy toward the VOCs, at least Delta and Omicron, since literature is available.
  • I would add more info regarding thrombosis and thrombocitopenia after receiving the first dose of AstraZeneca Vaccine. This was a game changing event for vaccination strategies, at least in Europe and needs to be addressed. By the way the correct name is "Schultz" not "Schult", fix this.

Minor points:

-Figure 1: change the scale considering infection and recovered each 100.000 inhabitants and deaths each million, it will deliver a more realistic picture.

-Line 128, put a comma between “subunits” and “Inactivated”

-Line 190, remove "vaccines"

-Line 292-293, what this paragraph regarding the placebo means? Please remove it

Line 304, put a brief explanation of Th2-skewed immunity

Line 359, remove doctor

Author Response

We thank the reviewers for their valuable comments and suggestions to improve the manuscript. We have addressed the comments and revised the manuscript according to the comments from the reviewers.

  1. Authors correctly highlighted the role of antioxidants in the frame of a possible COVID19 treatment without citing some interesting small molecules that are in the pipeline and some others that are in preclinical studies. In the first case Ebselen, the most studied among selenorganic compounds, needs to be cited since it is in phase III of clinical trials (https://clinicaltrials.gov/ct2/show/NCT04484025?term=ebselen&draw=2&rank=10; https://clinicaltrials.gov/ct2/show/NCT04483973?term=ebselen&draw=3&rank=11) this compound needs brief emphasis by citing recent and relevant literature. Here is some suggestions 10.1038/s41467-021-23313-7; 10.1038/s41586-020-2223-y and some recently reported review that authors can select form literature
    Authors cited quercetin as potential anti-SARSCoV2 agents, without citing related literature, here are some suggestions: 10.1016/j.ijbiomac.2020.07.235; 10.3390/ijms22137048; 10.3390/molecules26196062

We thank the reviewer for the comments and suggestions to improve the manuscript.

Response: Thank you for the comment. We have addressed the comment by including “Ebselen (2-phenyl-1,2-benzoisoselenazol-3(2H)-one), an organoselenium compound with hydroperoxide- and peroxynitrite-reducing activity acts by inhibiting the main viral protease, Mpro. It has proven potential beneficial effects of against COVID-19” and also the required references.

  1. Table 1. It is really interesting and well-conceived. I would remove the storage since it is not a useful information for readers aside of health authorities who practically will store such vaccines. Rather, I would add information regarding the efficacy toward the VOCs, at least Delta and Omicron, since literature is available.

Response: Thank you for the suggestion. We have addressed the comment by adding the information regarding the efficiency towards variants in table 1.

  1. I would add more info regarding thrombosis and thrombocitopenia after receiving the first dose of AstraZeneca Vaccine. This was a game changing event for vaccination strategies, at least in Europe and needs to be addressed. By the way the correct name is "Schultz" not "Schult", fix this.

Response: We appreciate the reviewer for giving us this suggestion to improve our manuscript. For addressing this comment, we have included “The five health care workers aged 32 to 54 years revealed venous thrombosis and thrombocytopenia 7 to 10 days after the initial dose. This shows that the thrombocytopenia was caused by a rare vaccine-related type of spontaneous heparin-induced thrombocytopenia.”
“Schultz” is also fixed.

Minor points:

-Figure 1: change the scale considering infection and recovered each 100.000 inhabitants and deaths each million, it will deliver a more realistic picture.
Response: Thank you for the comment. We have altered the picture according to the suggestion.

-Line 128, put a comma between “subunits” and “Inactivated”
 Response: Thank you for the comment. Line 134, “subunits, inactivated”. Comma added.

-Line 190, remove "vaccines"
Response: Thank you for the comment. Line 195, “vaccine”.

-Line 292-293, what this paragraph regarding the placebo means? Please remove it
Response: Thank you for the comment. Line 300, the sentence has been removed as per the suggestion.

Line 304, put a brief explanation of Th2-skewed immunity
Response: Thank you for the comment. For addressing this comment, line 315 “Since balanced T-cell response is required for an effective immune response, and immune response skewing has been associated to the emergence of pathologic responses, such as that seen in Respiratory syncytial virus (RSV) infection, which has enhanced Th2 and Th17 responses with attenuated the Th1 responses. Th2-skewed immune response is especially prominent in the lungs, resulting in severe bronchiolitis and asthma development, as well as respiratory dysfunction. Th2-skewed immunity, increase the risk of reinfection and induce the thickening of the epithelium”

Line 359, remove doctor

Response: Thank you for the comment. Line 377, “South African”. The word “Doctor” has been removed.

Round 2

Reviewer 3 Report

Authors correctly addressed all the requests, just a minor issue. When ebselen is detailed please cite also 10.3390/molecules26144230

Author Response

We thank the reviewer for their valuable comment and suggestion to improve the manuscript. We have addressed the comment and revised the manuscript according to the comments from the reviewer.

Reviewer 3:

  1. The authors correctly addressed all the requests, just a minor issue. When ebselen is detailed please cite also 10.3390/molecules26144230

Response: We thank the reviewer for suggesting this citation for improving our manuscript. We have addressed this comment and added this citation in Chapter 6 (Reference number- 150).